# Optical Absorption in N-Dimensional Colloidal Quantum Dot Arrays: Influence of Stoichiometry and Applications in Intermediate Band Solar Cells

**DOI:** 10.3390/nano12193387

**Published:** 2022-09-27

**Authors:** Rebeca V. H. Hahn, Salvador Rodríguez-Bolívar, Panagiotis Rodosthenous, Erik S. Skibinsky-Gitlin, Marco Califano, Francisco M. Gómez-Campos

**Affiliations:** 1Departamento de Electrónica y Tecnología de los Computadores, Facultad de Ciencias, Universidad de Granada, 18071 Granada, Spain; 2Pollard Institute, School of Electronic and Electrical Engineering, University of Leeds, Leeds LS2 9JT, UK

**Keywords:** colloidal quantum dots, optical absorption, ordered arrays, stoichiometry, dimensionality

## Abstract

We present a theoretical atomistic study of the optical properties of non-toxic InX (X = P, As, Sb) colloidal quantum dot arrays for application in photovoltaics. We focus on the electronic structure and optical absorption and on their dependence on array dimensionality and surface stoichiometry motivated by the rapid development of experimental techniques to achieve high periodicity and colloidal quantum dot characteristics. The homogeneous response of colloidal quantum dot arrays to different light polarizations is also investigated. Our results shed light on the optical behaviour of these novel multi-dimensional nanomaterials and identify some of them as ideal building blocks for intermediate band solar cells.

## 1. Introduction

In the search for ways of improving solar cell performance, the intermediate band solar cell (IBSC) is an intriguing concept that has been proposed to overcome the Shockley–Queisser limit [1], as part of the so-called “third generation devices”. The intermediate band (IB) is a narrow energy band placed within the bandgap having the role of a “stepping stone” for the electrons to promote from the valence band (VB) to the conduction band (CB), allowing the absorption of below-bandgap energy photons. Ever since the publication of the seminal paper by Luque and Martí [2], efforts in finding suitable materials or structures to achieve the formation of the IB have been pursued by many groups [3,4,5,6]. One of the most promising schemes that have been proposed so far is to use quantum dots to form the IB by coupling the states of the isolated dot into a miniband. Epitaxial quantum dots (EQDs) [7] have been previously investigated due to the possibility of creating one-dimensional (1D) vertical arrays of these nanostructures by stacking several layers of them. Strain in these systems favours the nucleation of new QDs at the same position in which a dot is present in the layer underneath, creating vertical 1D arrays through the layers. Unfortunately, however, the experimental performance of EQD-based IBSCs is far from satisfactory. In this regard, EQDs present two major problems [1]: weak absorption for transitions involving the IB, due to the low volumetric concentration of dots, and excessive nonradiative electron exchange between the IB and the valence (VB) or conduction band (CB) of the host, preventing voltage preservation at room temperature. Colloidal quantum dots (CQDs), on the other hand, have recently emerged as reliable candidates to overcome EQD limitations [5,8] as they can be densely packed, their size and shape is conveniently controllable [9] and they exhibit strong absorption in transitions involving the IB, when arranged in thin films [10]. The CQD synthesis processes are mature, and they make it possible to obtain samples with very low size dispersion, achieving good control over their properties [11,12] and to create ordered systems in 1D [13] (also known as linear CQD molecules, when a low number of quantum dots is used), 2D (as CQD films [14]) and 3D, with good spatial ordering [15]. On this basis, CQDs are promising candidates for the fabrication of cost-effective, highly efficient solar cells in diverse architectures such as Schottky junction quantum dot solar cells [16], quantum dot solar cells (QDSCs) based on a depleted heterojunction structure [17] and quantum junction solar cells [18]. Although further technological improvements are required to obtain well-ordered systems, the present state of the art is close to achieving these goals [5,10]. The possibility of achieving experimentally spatially ordered arrays, which behave similarly to regular crystalline solids, where CQDs play the role of regular atoms, has opened a new field of research intp these new systems. New physical phenomena associated to these structures, such as superradiance [19,20], are still intriguing and theoretical studies on them are of paramount importance. Quantum dot arrays also feature band-like carrier transport [21,22,23,24,25], which is important for improving carrier extraction from the solar cells, and provides the capability to engineer the array to target light absorption in specific photon energy ranges. As a result, CQD systems may have the features that are expected for an efficient absorber in IBSCs. However, it is still unclear which CQD arrangement (i.e., 3D, 2D or 1D) would perform better as the active region of such a device. Theoretical modelling is, therefore, needed to guide experiments in order to predict the advantages and drawbacks of these different systems, analysing the wide range of tunable parameters they offer: material, size, stoichiometry, interdot distance, and especially dimensionality (1D, 2D and 3D) can greatly influence the performance of a QD array in a solar cell. This work tries to shed light on the effect of some of these parameters on photon absorption. We studied arrays of spherical CQDs with the zinc-blended crystal structure, made of different In-based materials (InX, with X = P, As, Sb), and separated by one bond length. Although the dots contain all the same number of atoms, their radii differ slightly in different materials due to the difference in their lattice constant: R=11.9 Å for InP, R=12.2 Å for InAs and R=13.1 Å in the case of InSb. All these values are in perfect agreement with experimentally achieved CQD diameters [26,27,28]. The interdot separation is, in principle, limited by the presence of bulky passivants on the CQD surface. Many CQD solids can be placed within the first and second generations of assemblies. The former encompasses the assemblies formed by slow evaporation of the CQD dispersion, with long organic ligands [29]. To the latter belong the solids where the CQDs are assembled with short ligands [30]. In order to reduce the interdot distance, the aforementioned bulky passivants can be exchanged for shorter ones [14,31] or removed via thermal annealing [32,33]. The length of the capping agents most commonly used to stabilize the surface of the nanostructures ranges from 2 nm for oleic acid [34] to 0.35 nm for oxalic acid [35]. Inorganic ligands, such as atomic halide anions, can lead to a further decrease of the interdot separation down to 0.1 nm [36]. Recently, a third generation of CQD solids has emerged, where an oriented attachment of the CQDs is achieved by selectively removing the long ligand molecules from the surfaces, which promotes the possibility of direct attachment [25]. This third generation has the potential to realize strong electronic coupling of the QDs in a long-range-order assembly. All this experimental evidence supports our choice of one bond length (>0.2 nm) as an interdot separation which is actually experimentally feasible. The one-dimensional superlattice is constructed along the z-axis, the two-dimensional one is a square lattice extended in the xy-plane, and the three-dimensional array is a structure where the dots are periodically distributed along the x, y, z directions. The latter array consists of a system of stacked CQD films in which the dots are electrically linked between different vertical layers. In contrast, CQD films stacked with no electrical connection between layers (due to either a positional mismatch along the stacking direction, or a large interdot vertical distance), would behave as independent 2D arrays and would not exhibit the 3D behaviour discussed in this study. The materials investigated here were selected due to their reduced toxicity compared to other well-known Cd- and Pb-based CQD materials [37,38]. InP QDs have shown great potential in optoelectronic applications, InP/ZnS QDs (with or without metal doping) representing a promising material for white LEDs [39,40,41,42]. InAs is not a novel material for IBSCs, since arrays made of it and embedded into a GaAs matrix have been proposed previously for this application [6,7]. However, previous studies have modelled InAs dots as a continuous material by using **k·p** approaches, [3] and have therefore neglected atomistic details. The importance of considering such detail will be evident when we show the effects on both electronic structure and optical properties of inverting the stoichiometry of a CQD by interchanging anions and cations. We will also analyse the implications, for their application in IBSCs, of assembling CQDs with anion- and cation-rich surfaces.

As far as we know, such a comprehensive study on QD arrays, where different dimensionalities and stoichiometries are studied and compared, within the same theoretical framework, for three different materials, using different light polarization directions and for different temperatures, has not been conducted before. It should be considered that the direction of light polarization is crucial to ascertain the suitability of each combination of array dimensionality, material and specific stoichiometry for application in solar energy harvesting, where the incident light is unpolarized. It is also important to investigate how dimensionality influences the features of systems made of QDs of the same material, to identify the array configurations that are better suited to specific technological applications. Previous studies such as [3,7] have devoted effort to this topic in the last years, but none of them integrates a comparison between one-, two- and three-dimensional QD arrays. The effect of temperature on the system performances is also investigated, for completeness. Finally, the profound influence of the QD surface stoichiometry on the photochemical properties of these systems has been extensively proven experimentally: unlike simpler continuum-like approaches, such as k · p-based methods [3,7], our atomistic modelling is well suited to capture this effect as well.

## 2. Theoretical Framework

We first solve the Schödinger equation for an isolated quantum dot within the semiempirical pseudopotential method framework [43,44] using the folded spectrum technique [45]. The set of eigenstates, $\phi_{m}\left(\vec{r}\right)$, and eigenenergies we obtain are used as a basis to solve the following Schrödinger equation for the quantum dot array: (1)$$\left[T+\sum_{{\vec{R}}_{n}}V_{{\vec{R}}_{n}}\right]\psi_{\vec{q}}\left(\vec{r}\right)=E_{\vec{q}}\psi_{\vec{q}}\left(\vec{r}\right),$$
where T is the kinetic energy operator, ${\vec{R}}_{n}$ are the quantum dot positions in the array (in 1D, 2D and 3D), $V_{{\vec{R}}_{n}}$ is the potential of the quantum dot placed at ${\vec{R}}_{n}$, $\psi_{\vec{q}}\left(\vec{r}\right)$ is the eigenstate wavefunction, $E_{\vec{q}}$ is its associated eigenenergy and $\vec{q}$ is the wave vector in the array reciprocal space. The array potential is obtained as a summation of the isolated dot potentials placed at their positions in the array. The array wavefunctions are expanded as [46]: (2)$$\psi_{\vec{q}}\left(\vec{r}\right)=\frac{1}{\sqrt{N_{uc}K}}\sum_{m}\sum_{{\vec{R}}_{n}}b_{m}\exp\left(i\vec{q}\cdot {\vec{R}}_{n}\right)\phi_{m}\left(\vec{r}-{\vec{R}}_{n}\right)=\frac{1}{\sqrt{N_{uc}K}}\exp\left(i\vec{q}\cdot \vec{r}\right)u_{\vec{q}}\left(\vec{r}\right),$$
where $1/\sqrt{N_{uc}K}$ is the wavefunction normalization constant, $N_{uc}$ is the number of unit cells in the array, $b_{m}$ are the expansion coefficients, $\vec{q}$ is the array reciprocal space vector and $u_{\vec{q}}\left(\vec{r}\right)$ is the periodic function given by Bloch’s theorem. This is a general expression for the wavefunction regardless of the quantum dot dimensionality. The only implication of the dimensionality is in the reciprocal space: Although $\vec{r}$ is always a three-dimensional vector, the ${\vec{R}}_{n}$ positions form a 1D, 2D or 3D array from which a reciprocal space with the same dimensionality is derived. Therefore, the $\vec{q}$ vectors can belong to a 1D, 2D or 3D reciprocal space, accordingly. In this work, we investigate the effects of the array dimensionality on the photon absorption coefficient. In previous papers, we derived an expression for the photon absorption coefficient in quantum dot films within the electric dipole moment approximation using Fermi’s Golden Rule [47]: (3)$$\alpha =\frac{2\pi e^{2}}{Q_{st}v_{uc}n_{r}c\epsilon_{0}\Delta E}\sum_{i}\sum_{f}\frac{\omega_{if}}{K_{i}K_{f}}|\langle u_{f}|\hat{e}\cdot \vec{r}|u_{i}{\rangle_{uc}|}^{2}\left(f\left(E_{i}\right)-f\left(E_{f}\right)\right),$$
where *e* is the electron charge, $Q_{st}$ is the number of vectors of the reciprocal space for which the Schrödinger equation is solved (a 51 grid discretization per reciprocal space axis is used here to sample the Brillouin zone, i.e., 51 in 1D, 51×51 in 2D and 51×51×51 in 3D), $v_{uc}$ is the volume of the superlattice unit cell, $n_{r}$ is the refractive index of the material (for simplicity, here we use $n_{r}=1$), *c* is the speed of light in vacuum, $\epsilon_{0}$ is the vacuum dielectric constant, ΔE is the interval width within which energy is assumed to be conserved, (i.e., the Dirac’s delta function is approximated as a window function of constant value 1/ΔE), ω is the angular frequency of the photon involved in the absorption process, $\hat{e}$ is the unit vector along the electromagnetic wave potential vector (the radiation wave front is perpendicular to this vector), $u_{f}$ and $u_{i}$ are the Bloch functions of the superlattice wavefunction for the final (subscript *f*) and initial (subscript *i*) states, and f(E) is the Fermi–Dirac’s statistics. This expression could be extended to other periodic quantum dot arrangements. The only quantity that should be clarified is the unit cell volume. In a 3D quantum dot array this quantity is well defined by the interdot distances. In a 2D film, the unit cell area is well defined, but the film thickness is considered in this work to be the quantum dot diameter. The same holds for a 1D array, where the unit cell is well defined along the array axis while the cross-section is not properly defined. We used a square with the side length equal to the quantum dot diameter. Other choices for these lengths would modify the final absorption values by a multiplicative factor, which would have an impact neither on the calculated absorbed photon energies, nor on the relative absorption intensities between different minibands.

## 3. Results and Discussion

We analysed arrays of three different III-V In-based binary QDs (InX, X = P, As, Sb) in order to elucidate the influence of the anion atomic number and the array dimensionality on the photon absorption, Figure 1 shows a schematic representation of the CQD arrays implemented in one, two and three dimensions. For each dimensionality, we investigated the impact of stoichiometry, light polarization and temperature on the absorption properties. We consider spherical quantum dots of two stoichiometries: anion-rich surface (configuration A), and cation-rich surface (configuration B), obtained by inverting the stoichiometry of configuration A. Indeed, it has been shown experimentally that it is possible, by using specific capping groups, to control the surface composition in II–VI and IV–VI CQDs, thereby affecting the stability of the dots and their optical properties [48,49]. In more covalent compounds such as III–V CQDs, configuration B has already been achieved experimentally [50,51,52], whereas configuration A is more challenging due to the existence of fractional dangling bonds on the CQD surface. In this study, both configurations have been taken into account for the sake of completeness but, as will be shown later in this paper, configuration A is not the most adequate for its implementation in IBSCs. As the lowest energy conduction miniband (the CBM) has the potential to play the role of the IB in the active region of a quantum dot solar cell, we fixed the position of the Fermi level in the middle of that miniband in each case, guaranteeing that the miniband will be populated with electrons to promote to the upper minibands, and to also have empty states to accommodate the electrons promoted from the lower energy (i.e., valence) minibands.

### 3.1. Electronic Structure

When displaying the miniband structure of the different configurations (Figure 2, Figure 3 and Figure 4), a representative path in the Brillouin zone has to be selected. These paths are shown as insets in Figure 3 and Figure 4. We find that the change in the dot stoichiometry induces a noticeable variation in the energy spectrum, particularly in the conduction minibands. Such a strong dependence of the array optical properties, crystalline structure and stability on surface composition was also previously observed experimentally [48,53,54]. Arrays of InAs and InSb QDs with cation-rich surfaces seem to be suitable candidates as active materials in an IBSC, given that the energy separation between IB and CB (∼0.7 eV and ∼0.45 eV at Γ, respectively) and between VB and IB (∼1.75 eV and ∼1.56 eV at Γ, respectively) are close to the ideal conditions of about 0.7 eV and 1.23 eV already reported [2]. Similar arguments seem to preclude the use of InP CQDs (∼0.26 eV and ∼2.29 eV for the IB-to-CB and for the VB-to-IB gaps, respectively) in IBSCs. As will be discussed later, this material shows particular features that could represent a major drawback to this specific technological application. From Figure 2 we see that in the one-dimensional case, and especially for CQDs with a cation-rich surface, the CBM is perfectly separated from the bands located higher in energy (denoted as CBM + *n*, n=1,2,…). It is also clear that arrays of dots with cation-rich surfaces (configuration B) exhibit flatter minibands than arrays with anion-rich surfaces (configuration A). This is a consequence of the stronger interdot coupling occurring in configuration A (i.e., to the stronger interaction taking place between anion-rich surfaces compared to that between cation-rich ones). This effect is shown quantitatively in Table 1, where the right column is related to the miniband width (whose values are greater in configuration A than in configuration B, and of similar magnitude for all anion-rich surfaces), whereas the left column provides an estimate of the wavefunction confinement (which is higher for configuration B than it is for configuration A). This effect is observable not only in the 1D case, but also in 2D and 3D structures.

Figure 3 shows the miniband structure calculated for 2D arrays of In-based CQDs, with both anion-rich (configuration A) and cation-rich surfaces (configuration B). As in the 1D case, flatter minibands are obtained for the latter. The width of the minibands is greater than in the 1D case, due to the increased number of neighbours in the array, leading to a greater wavefunction overlap. It is worth noting that, for configuration A, the gap between the CBM and the CBM+1 decreases near the boundaries of the Brillouin zone. As will be shown later, this reduction crucially influences the photon absorption profile, allowing the absorption of photons with lower energies.

Following the trend of the 1D and 2D cases, the minibands are wider in the 3D case, as it can be seen in Figure 4. The gap between CBM and CBM+1 is nearly closed along the M-R line in arrays of dots with anion-rich surfaces, nearly removing the main IB features in these configurations. This suggests that 3D arrays made of QDs with anion-rich surfaces might be less suitable as active regions in IBSCs than those with cation-rich surfaces, at least with the configuration studied within this work.

### 3.2. Absorption Coefficient

Figure 5, Figure 6 and Figure 7 show the absorption coefficients calculated at 300 K for 1D, 2D, and 3D CQD arrays made of InP, InAs and InSb, respectively, with both anion-rich (left columns) and cation-rich (right columns) surfaces. In all cases, we consider three different light polarizations: along the *x* (100), *y* (010) and *z* (001) axes. The latter corresponds to normal illumination to the array axis in the 1D case, and to tangential illumination in the 2D case. Absorption in three distinct energy ranges is observed for configurations A and B in InAs and InSb and for configuration A in InP. Given that we are considering three separate transitions (from the valence band to the intermediate band, VB → IB, from the intermediate band to the conduction band, IB → CB, and from the valence band to the conduction band, VB → CB) this feature is expected, and has also been reported elsewhere [3,6], where an energetically isolated IB is observed.

In the case of arrays of InAs dots with cation-rich surfaces, Figure 6b,d,f shows that the absorption range for the IB → CB transition lies between ∼0.6 eV and 0.8 eV, that for the VB → IB transition between ∼1.9 eV and 2.2 eV, and that for the VB → CB transition starts at about 2.6 eV. In contrast, in the case of anion-rich surfaces (panels Figure 6a,c,e), the different transitions are not always as clearly distinguishable. This is a consequence of the reduced miniband gap near the Brillouin zone boundaries.

The behaviour is similar in arrays of InSb CQD (see Figure 7). An interesting peculiarity arises in InP CQD arrays with cation-rich surfaces (Figure 5b,d,f): Following the trend of InAs and InSb, the presence of three distinct photon energy ranges was expected. However, absorptions from the CBM are absent in InP arrays, as well as most of the transitions arriving to it, due to the existence of a “selection rule” for configuration B. As can be seen from Appendix A, transitions from VBM-8 to CBM and from VBM-7 to CBM show the greatest value of the oscillator strength. It is necessary to mention here that the oscillator strength does not take into account the occupation of the involved states. If the final state is fully occupied, the transition will not occur, although the oscillator strength is not negligible. Therefore, the former absorption transition is blocked due to the CBM being half-filled, Appendix A. The latter, on the contrary, is allowed by the population of the CBM and it is observable in the absorption profile of Figure 5. The energy of this transition is about 2.62 eV. This fact can be attributed to the number of P atoms at the surface of the QD, which has been found to impact its optical properties [55], given that in contrast to As and Sb, the P atom does not have any occupied d-orbital, bringing about a peculiar electronic structure. Configuration A, however, displays the same behaviour as its counterparts in InAs and InSb. We performed a reciprocal space decomposition of the isolated QD CBM+n wavefunctions. The results are depicted in Figure 8 and Figure 9. These decompositions were performed using a Voronoi partition of the Brillouin zone. The terms of the decomposition are added depending on the closest high-symmetry point, as described in [56]. From that, we can gain an insight of the contribution of low and high frequencies in the wavefuctions. Figure 8 and Figure 9 suggest that the CBM+*n* wavefunctions in arrays of InP with a cation-rich surface have higher frequency components in the spectrum than those of InAs because of the higher contributions around L and X points in the Brillouin zone. These oscillations would imply high frequency components in the products $\psi_{CBM}^{*}\left(\vec{r}\right)\psi_{CBM+n}\left(\vec{r}\right)$, explaining the observed behaviour in absorption. Appendix A illustrates this fact. As the number of oscillations increases in the product of the wavefunctions, the positive and negative contributions in the integrand of the oscillator strength integral become comparable, making the oscillator strength integral tend to zero. In contrast, in arrays of InP, InAs and InSb CQDs with anion-rich surfaces, the wavefunctions exhibit similar contributions from Γ, L and X, which would guarantee the presence of the three different energy ranges in absorption. As dimensionality increases, the energy ranges relative to the different transitions widen and greatly overlap, which has also been reported elsewhere [6], due to the higher curvature of the minibands. This is a drawback to the use of 3D arrays in IBSCs, given that the miniband bending results in higher energy losses via thermalization. When the dots are located one bond length apart, the absorption coefficients calculated for 3D arrays of CQDs with anion-rich surfaces overlap, as shown in Figure 5, Figure 6 and Figure 7. This leads to a reduction in the IBSC efficiency under fully concentrated sunlight [57]. However, it has been demonstrated [58] that the same partial overlapping condition, under unconcentrated sunlight, leads to an increased efficiency in IBSCs with energy gaps between 0.5 eV and 2.7 eV. We, therefore, conclude that arrays of dots with anion-rich surfaces would be ideal for operation under unconcentrated sunlight, whereas dots with cation-rich surfaces would be better suited for fully concentrated light conditions. For all array dimensions, the absorption coefficient appears to be rather independent of temperature, slightly decreasing with it as discussed elsewhere [10,59]. It is straightforward to conclude that the dependence on temperature of the light absorption coefficient is small, as it is only included in the term $f\left(E_{i}\right)-f\left(E_{f}\right)$ of Equation (Equation 3). With respect to the effect of the Fermi level position in the aforementioned coefficient, taking again as basis Equation (Equation 3), if $E_{i}$ and $E_{f}$ are fixed and $E_{F}$ is allowed to vary between these two extreme values, the factor $f\left(E_{i}\right)-f\left(E_{f}\right)$ ranges between 1 and 0.5. We find that the light polarization has, instead, a strong influence on the absorption profile, as also mentioned elsewhere [60]. This influence, however, varies with dimensionality and surface stoichiometry. For example, light absorption in arrays with cation-rich surfaces is nearly independent of light polarization, the only remarkable exceptions being (i) the strongly suppressed absorption for (100) polarization in InP, which makes arrays of any dimensionality almost transparent for light polarized in that direction, and (ii) the higher absorption of (001)-polarized light (i.e., for tangent incidence) in InAs films for IB→BC transitions, while (010) polarization is favoured for VB→IB and VB→CB transitions. The dependence on light polarization is more accentuated in arrays of CQDs with anion-rich surfaces: IB→CB transitions are stronger for (001)-polarized light in films of InP and InSb dots, while this polarization is more weakly absorbed for VB→IB and VB→CB transitions in 2D arrays of InAs and InSb CQDs. It is important to point out that polarization-independent absorption represents a huge advantage for application in solar cells.

## 4. Conclusions

We conducted an investigation into the influence of dimensionality and surface stoichiometry on the electronic structure and photon absorption in arrays of InP, InAs and InSb CQDs containing the same total number of atoms, for applications in IBSCs. Our results show that dimensionality plays a crucial role in optimizing CQD-based arrays for IBSCs. Indeed, they indicate that 3D CQD arrays may degrade the IB features due to the miniband widening caused by the increased interdot interaction occurring in CQD solids compared to 2D and 1D arrays. Our results, therefore, suggest that stacking suitably spaced 2D arrays could result in improved IBSCs performances, compared with the use of 3D arrays, considering that they will provide the same optical thickness for the active region without, however, exhibiting interlayer coupling along the stacking direction. We also found that the nature of the atomic species on the QD surface has a dramatic effect on both the electronic miniband structure and the optical properties of the array. In particular, the absorption features of the latter can be varied from overlapping (using anion-rich surfaces) to non-overlapping (with cation-rich surfaces), either of which, when used in IBSCs, can be suitable in different illumination conditions. To help guide the experimental work on these materials, we, therefore, recommend the use of closely packed InAs or InSb CQD films with cation-rich surfaces for applications under fully concentrated sunlight. Finally, our findings show that CQD arrays are ideally suited to be used in the active region of IBSCs, since their absorption properties are nearly polarization-independent. This is an uncommon feature of these systems, especially desirable considering the unpolarized nature of solar radiation.

## Figures and Tables

**Figure 1 nanomaterials-12-03387-f001:**
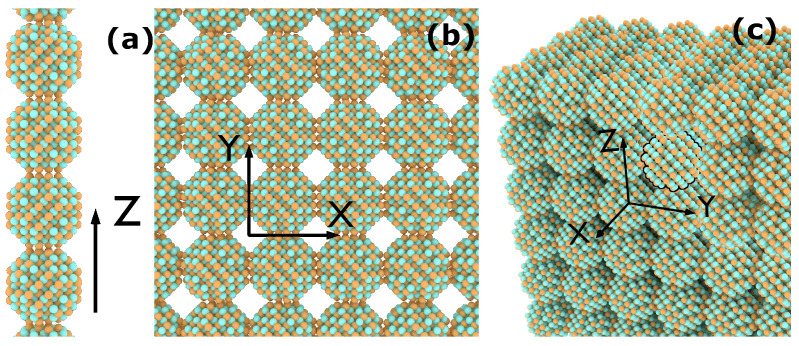
Schematic representation of the simulated CQD arrays in (**a**) 1D, (**b**) 2D and (**c**) 3D. For the sake of clarity, a single quantum dot is highlighted in the 3D structure.

**Figure 2 nanomaterials-12-03387-f002:**
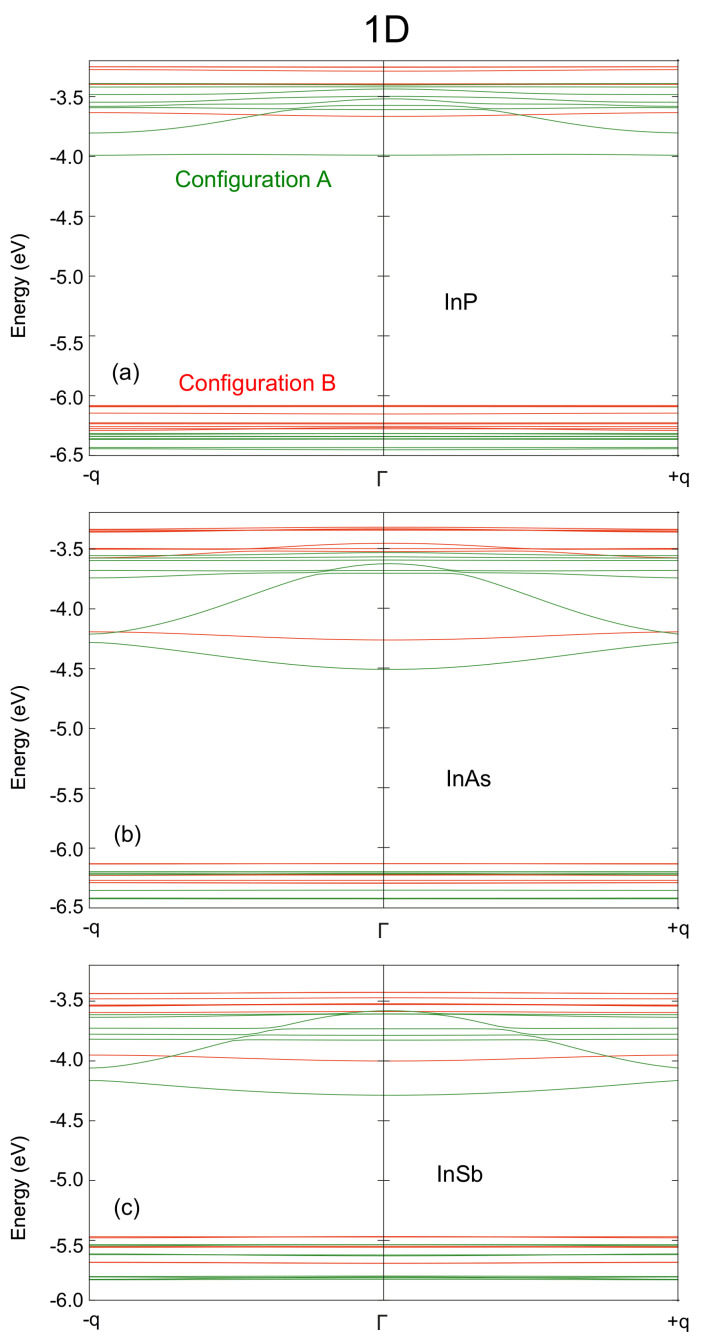
Miniband structure for 1D arrays of (**a**) InP, (**b**) InAs, and (**c**) InSb QDs with anion-rich (configuration A, green lines) and cation-rich (configuration B, red lines) surfaces. The energies on the vertical axis are relative to the vacuum level; the horizontal axis sweeps the Brillouin zone, q stands for π/2$R_{1}$, i.e., the limits of the Brillouin zone.

**Figure 3 nanomaterials-12-03387-f003:**
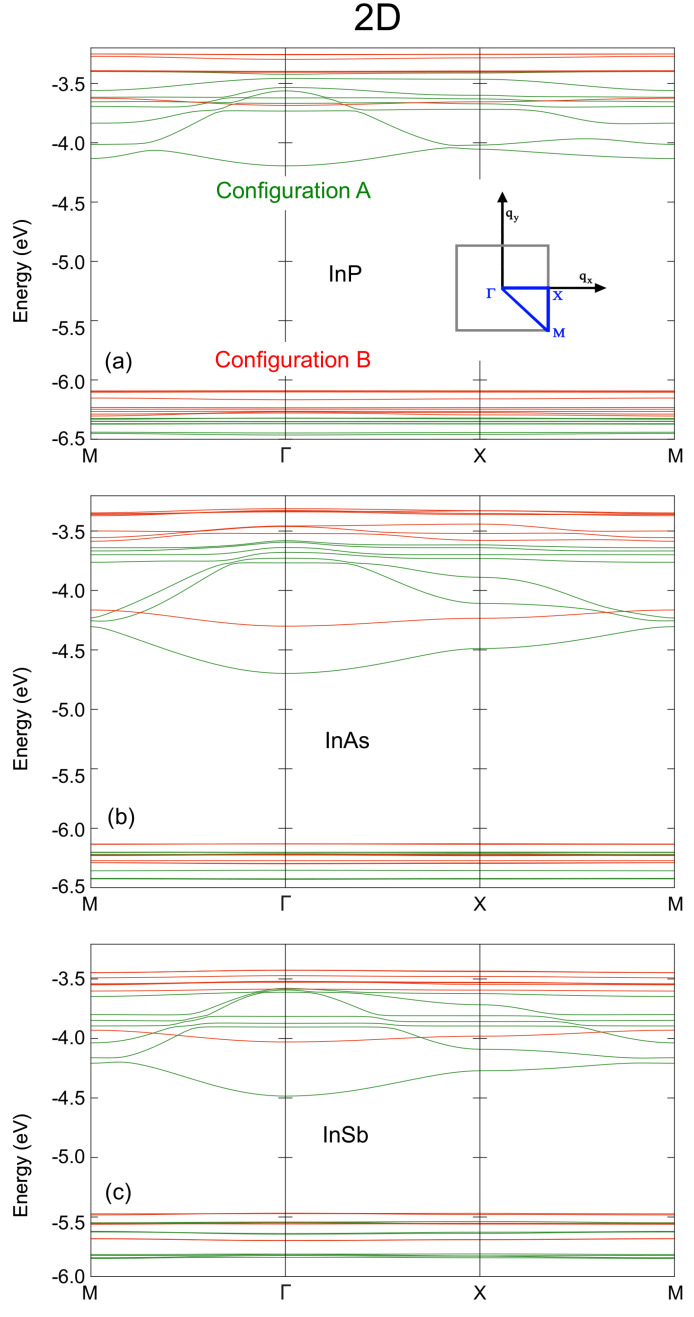
Miniband structure for 2D arrays of (**a**) InP, (**b**) InAs, and (**c**) InSb QDs with anion-rich (configuration A, green lines) and cation-rich (configuration B, red lines) surfaces. The energies on the vertical axis are relative to the vacuum level; the horizontal axis sweeps the Brillouin zone. The inset in (**a**) shows a schematic view of the path along the Brillouin zone together with the indexing of its major points.

**Figure 4 nanomaterials-12-03387-f004:**
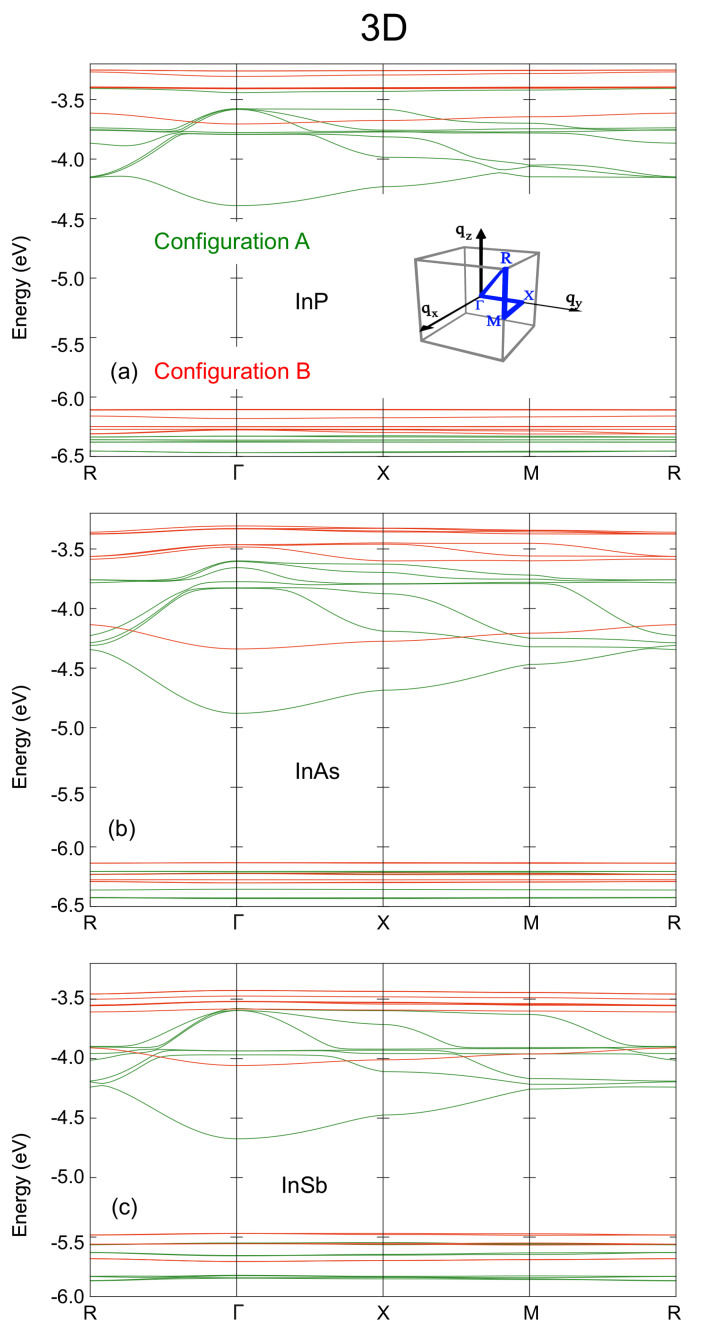
Miniband structure for 3D arrays of (**a**) InP, (**b**) InAs, and (**c**) InSb QDs with anion-rich (configuration A, green lines) and cation-rich (configuration B, red lines) surfaces. The energies on the vertical axis are relative to the vacuum level; the horizontal axis sweeps the Brillouin zone. The inset in (**a**) shows a schematic view of the path along the Brillouin zone together with the indexing of its major points.

**Figure 5 nanomaterials-12-03387-f005:**
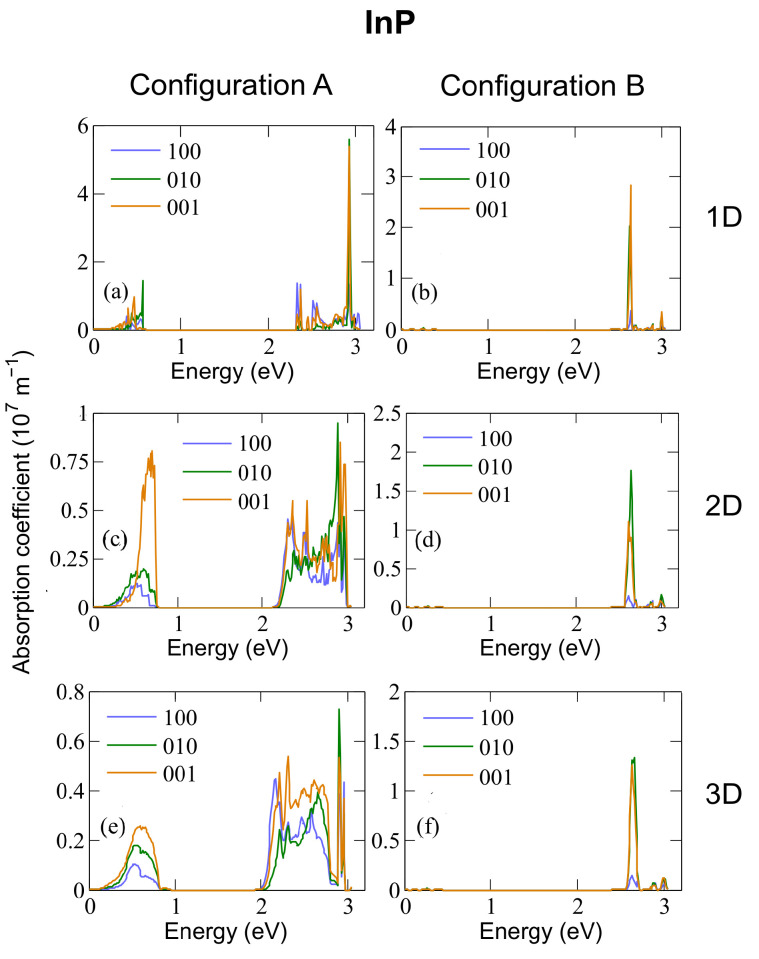
Absorption coefficient calculated at 300 K for 1D (**a**,**b**), 2D (**c**,**d**) and 3D (**e**,**f**) arrays of InP dots with radius R=11.9 Å and either anion-rich (configuration A, left panels) or cation-rich (configuration B, right panels) surfaces.

**Figure 6 nanomaterials-12-03387-f006:**
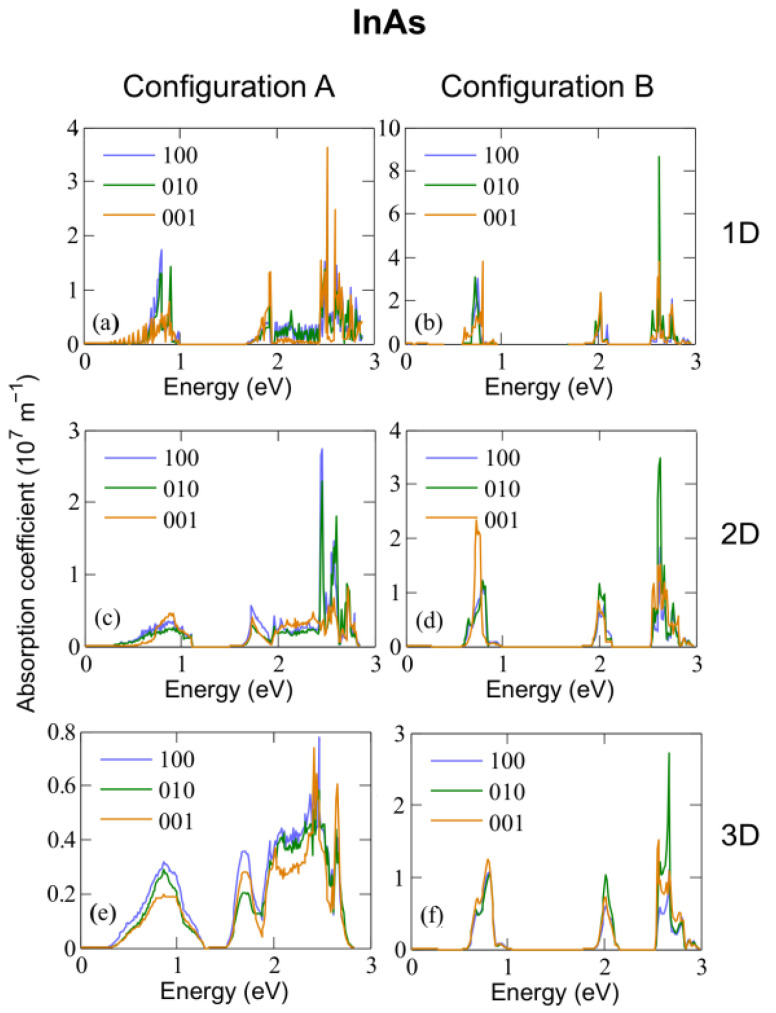
Absorption coefficient calculated at 300 K for 1D (**a**,**b**), 2D (**c**,**d**) and 3D (**e**,**f**) arrays of InAs dots with radius R=12.2 Å and either anion-rich (configuration A, left panels) or cation-rich (configuration B, right panels) surfaces.

**Figure 7 nanomaterials-12-03387-f007:**
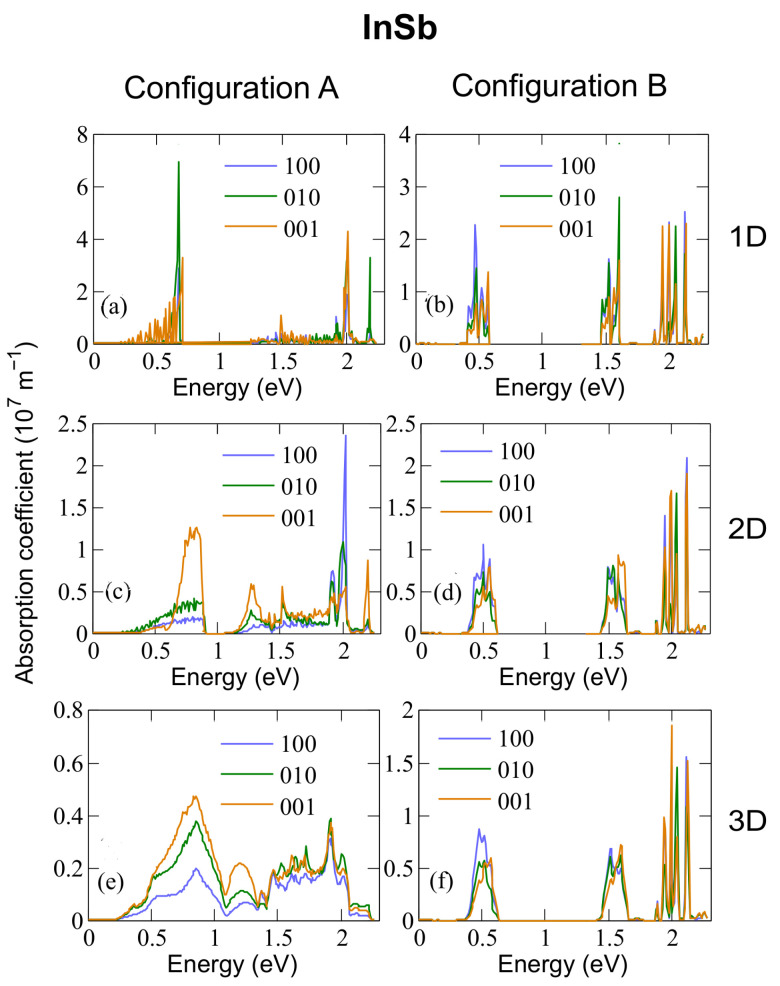
Absorption coefficient calculated at 300 K for 1D (**a**,**b**), 2D (**c**,**d**) and 3D (**e**,**f**) arrays of InSb dots with radius R=13.1 Å and either anion-rich (configuration A, left panels) or cation-rich (configuration B, right panels) surfaces.

**Figure 8 nanomaterials-12-03387-f008:**
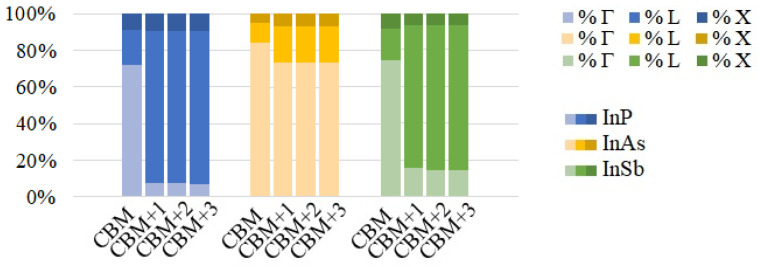
High symmetry points contribution to the CB states of InP, InAs and InSb QDs with anion-rich surfaces.

**Figure 9 nanomaterials-12-03387-f009:**
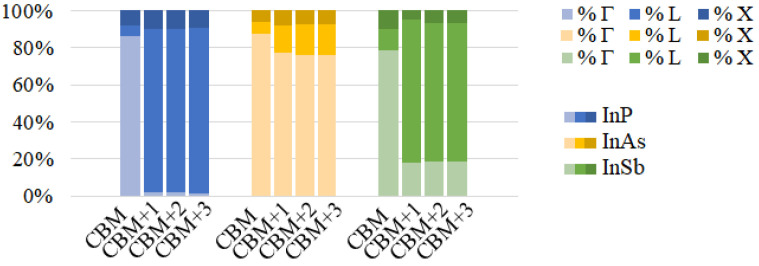
High symmetry points contribution to the CB states of InP, InAs and InSb QDs with cation-rich surfaces.

**Table 1 nanomaterials-12-03387-t001:** Overlap integrals between neighbouring QDs for CBM states. The integral on the left quantifies the confinement of the wavefunction in the QD, whereas the integral on the right is related to the miniband width.

Overlap Integrals			
		$\int \psi^{*}\left(\vec{r}\right)\psi \left(\vec{r}-\vec{R}\right)d\vec{r}$ (adim.)	$\int \psi^{*}\left(\vec{r}\right)V\left(\vec{r}\right)\psi \left(\vec{r}-\vec{R}\right)d\vec{r}$ (eV)
	InP	0.0123	−0.0479
Configuration A	InAs	0.0141	−0.0571
	InSb	0.0146	−0.0599
	InP	0.0032	−0.0076
Configuration B	InAs	0.0066	−0.0171
	InSb	0.0058	−0.0125

## Data Availability

Data supporting the results presented in the paper are available upon reasonable request to the authors.

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
