# Peer review of "Optical Absorption in N-Dimensional Colloidal Quantum Dot Arrays: Influence of Stoichiometry and Applications in Intermediate Band Solar Cells"

_nanomaterials, 2022, doi:10.3390/nano12193387_

Round 1

Reviewer 1 Report

In this manuscript, the electronic structure and absorption coefficient of three different III-V In-based binary QDs (InX, X = P, As, Sb) and their dependence on array dimensionality and surface stoichiometry were discussed detailly. The research has a certain novelty and the article is also well written. It is recommended to be accepted after a minor revision.

1.      When studying the microstrip structure of multi-dimensional array, the relevant band gap at the high symmetry point should be marked to make the change of energy band more concrete.

2.      It is recommended to appropriately narrow the range of the ordinate in Figure 1-3 to reflect the separation of the energy bands more intuitively.

3.      It is suggested to add a schematic diagram of the crystal structure of colloidal quantum dots.

4.      Introduction, when discussing the synthesis and optical properties of In-based quantum dots, the following reference is relevant to this topic, which could help readers better understand the latest progress: Small, 2022, 18(15): 2108120; Langmuir, 2020, 36(34): 10244-10250; Applied Surface Science, 2019, 493: 605-612; Journal of Luminescence, 2019, 212: 264-270.

5.      Section 3 should be results and discussion, and Section 4 should be conclusion.

Reviewer 2 Report

In this paper, the authors have studied theoretically the optical properties of nontoxic INX (x = P, as, sb) colloidal quantum dot arrays for photovoltaic applications. Here, the authors focus on the electronic structure and light absorption, as well as their dependence on array dimension and surface stoichiometry. The uniform response of colloidal quantum dot arrays to different polarizations was also studied. The results of this paper reveal the optical behavior of these new multi-dimensional nano materials, and confirm that some of them are ideal building blocks for intermediate band solar cells. I believe that publication of the manuscript may be considered only after the following issues have been resolved.

1.      What are the advantages of this job? It is suggested that the author give a table to compare the advantages of this work in detail.

2.      The author needs to give the schematic diagram of the model in the paper.

3.      Figure 7 and figure 8 are not explained in the article by the author and need to be supplemented in the article.

4.      The introduction can be improved. Some works on solar cells and theirs related properties should be added such as Physical Chemistry Chemical Physics, 2022, 24, 4871 – 4880. Some works on quantum dots and theirs related properties should be added such as RSC Adv. 9 (2019) 41383–41391; RSC Adv. 8 (2018) 42233–42245; Talanta 188 (2018) 145–151.

5.      The author of the discussion part needs to be concise and organized into a paragraph.

Round 2

Reviewer 2 Report

The authors have not responded well to solve the relevant problems, and the suggested article is unacceptable.
